

# Small herbivores and abiotic heterogeneity promote trait variation of a saltmarsh plant in local communities

Qingqing Chen[1,2], Christian Smit[1], Ido Pen[1] and Han Olff[1]

[1] Groningen Institute for Evolutionary Life Sciences (GELIFES), University of Groningen, Groningen, The Netherlands
[2] Current affiliation: Institute of Ecology, College of Urban and Environmental Science, Peking University, Beijing, China

## ABSTRACT

Intraspecific trait variation (ITV) enables plants to respond to global changes. However, causes for ITV, especially from biotic components such as herbivory, are not well understood. We explored whether small vertebrate herbivores (hares and geese) impact ITV of a dominant clonal plant (*Elytrigia atherica*) in local communities. Moreover, we looked at the relative importance of their direct (*e.g.,* selective grazing) and indirect effects (altering genotypic richness/diversity and abiotic environment) on ITV. We used exclosures at two successional stages in a Dutch saltmarsh, where grazing pressure at the early successional stage was ca. 1.5 times higher than that of the intermediate successional stage. We measured key functional traits of *E. atherica* including height, aboveground biomass, flowering (flower or not), specific leaf area, and leaf dry matter content in local communities (1 m × 1 m plots) inside and outside the exclosures. We determined genotypic richness and diversity of each plant using molecular markers. We further measured abiotic variations in topography and clay thickness (a proxy for soil total nitrogen). Structural equation models revealed that small herbivores significantly promoted ITV in height and flowering at the early successional stage, while they marginally promoted ITV in height at the intermediate successional stage. Moreover, the direct effects of herbivores played a major role in promoting ITV. Small herbivores decreased genotypic diversity at the intermediate successional stage, but genotypic richness and diversity did not impact ITV. Small herbivores did not alter topographic variation and variation in clay thickness, but these variations increased ITV in all traits at the early successional stage. Small herbivores may not only impact trait means in plants as studies have shown but also their ITV.

Corresponding author
Qingqing Chen,
chqq365@hotmail.com

## INTRODUCTION

Studies show that intraspecific trait variation (*i.e.,* variability in traits of conspecific individuals; hereafter ITV) can enable plant species to respond to global changes (*Westerband, Funk & Barton, 2021*), impact community composition and structure (*Whitlock, Grime & Burke, 2010*), and govern ecosystem processes (*Boege & Dirzo, 2004*;

*Crutsinger et al., 2006*; *Crutsinger, Souza & Sanders, 2008*; *Lecerf & Chauvet, 2008*). The importance of ITV has been increasingly realized over the years (*Violle et al., 2012*; *Siefert et al., 2015*; *Funk et al., 2017*; *Westneat et al., 2019*; *Westerband, Funk & Barton, 2021*), therefore, exploring the causes for ITV is of prime importance, particularly under the current rapid global changes (*Westerband, Funk & Barton, 2021*). Several components including genotypic richness/diversity and phenotypic plasticity triggered by biotic and abiotic environments can drive ITV (*Valladares, Gianoli & Gomez, 2007*). However, biotic components, particularly herbivory, received much less attention than genotypic richness/diversity and abiotic environments (*Valladares, Gianoli & Gomez, 2007*).

Whilst herbivores are one of the major drivers for plant trait differentiation globally (*Díaz et al., 2007*; *He & Silliman, 2016*), the majority of the studies that document the effects of herbivores on plant traits focus on trait means without considering ITV (*e.g.*, *Bullock et al., 2001*; *Louault et al., 2005*; *Kahmen & Poschlod, 2008*). Limited studies suggest that herbivores may also impact ITV (*e.g.*, *Jessen et al., 2020*). However, these studies mainly focus on large herbivores. Large and small herbivores are relative. Here we refer to small herbivores as vertebrate herbivores with body mass ranging from 1 to 10 kg such as hares and geese. Small herbivores sometimes can have stronger impacts on plant communities than large ones especially when their abundance is high (*Olofsson et al., 2004*). Additionally, small herbivores are usually more selective with forage plants (*Olff & Ritchie, 1998*). Therefore, small herbivores may also impact ITV, however, we currently lack empirical evidence. As small herbivore populations are changing dramatically due to human actions (*Smith, Jennings & Harris, 2005*; *Menu, Gauthier & Reed, 2002*), it is important to understand whether and how small herbivores impact ITV.

Small herbivores may impact ITV *via* multiple non-exclusive processes. First, small herbivores can promote ITV through selective grazing. For instance, studies show that large herbivores generally prefer young plants that are often more nutritious (*e.g.*, higher protein and lower lignin and cellulose) (*Augustine & McNaughton, 1998*). Thus, larger herbivores decrease plant height by grazing them down (*Díaz et al., 2007*; *Evju et al., 2009*). Similarly, small herbivores may prefer young plants and decrease plant height, but because small herbivores consume less and they are more selective with forage plants, some young plants may escape from small herbivores *via* association with unpalatable species (*Van Der Wal et al., 2000b*; *Kuijper & Bakker, 2008*). Thus, small herbivores may promote variation in plant height. Second, small herbivores may impact ITV indirectly through altering genotypic richness and diversity. Increased genotypic richness and diversity can increase ITV (*Evans et al., 2016*). Studies looking at the effects of large herbivores on genotypic diversity show that large herbivores can increase genotypic diversity by promoting seed transportation (*Rico & Wagner, 2016*) or decrease it by promoting clonal growth (*Kleijn & Steinger, 2002*). Small herbivores can have similar effects to large ones on many processes such as seed dispersal (*Bakker & Olff, 2003*). Therefore, small herbivores can either increase or decrease genotypic richness and diversity. Third, small herbivores may promote ITV indirectly through altering abiotic heterogeneity (*e.g.*, topographic variation and variation in soil nutrients) *via* grazing the vegetation at particular spatial scales, trampling, and localized deposition of droppings (*Adler, Raff & Lauenroth, 2001*).

Abiotic heterogeneity usually promotes ITV (*Westerband, Funk & Barton, 2021*). However, whether small herbivores impact ITV *via* these processes and the relative importance of them in shaping ITV remains underexplored.

To test these hypotheses, we used long-term (22-year) exclosure experiment at the early and intermediate successional stages of a Dutch saltmarsh. In the eastern part of the saltmarsh of the island of Schiermonnikoog, hares and geese are the abundant herbivores while large herbivores are absent (*Kuijper & Bakker, 2005*; *Schrama et al., 2015*; *Chen et al., 2019b*). A well-calibrated successional gradient is present here (*Olff et al., 1997*). Hares and geese are more abundant at the early successional stages but their abundance decreases at intermediate and late successional stages. This is because less preferred late successional plants such as *Elytrigia atherica* (synonym *Elymus athericus*); a perennial grass increase in dominance (*Kuijper & Bakker, 2005*; *Schrama et al., 2015b*; *Chen et al., 2019*). *Elytrigia atherica* mainly reproduces clonally, but sexual reproduction may occur occasionally *via* windows of opportunity and their seeds are mainly dispersed by tidal water, thus genotypic richness is high in this plant (*Bockelmann et al., 2003*; *Chang et al., 2005*; *Chen, 2020*). Topographic variation and variation in clay thickness are important factors for the growth and expansion of *E. atherica* in saltmarshes (*Olff et al., 1997*; *Nolte et al., 2019*). Thus, the long-term exclosure experiment along the successional gradient in this saltmarsh provides ideal conditions to look at the effects of small herbivores on ITV in this dominant plant and whether the effects are through different processes.

In this study, we look at the effects of hares and geese on ITV of *E. atherica* in local communities (1 m × 1 m plots). This is because ITV is usually strongly driven by processes operating at small spatial scales (*Westerband, Funk & Barton, 2021*). We measured key functional traits of this dominant plant including height, aboveground biomass, flowering (flower or not), specific leaf area, and leaf dry matter content within local communities inside and outside the exclosures. Using structural equation models, we test whether hares and geese impact ITV of *E. atherica*, and the relative importance of their direct effects (*e.g.*, selective grazing) and indirect effects (*via* altering genotypic richness/diversity and abiotic variables) on ITV. We expect that hares and geese would increase ITV in all traits measured and their effects would be more apparent at the early successional stage (relative to the intermediate one) where grazing pressure was higher. Although adult plants of *E. atherica* are less preferred, hares and geese considerably graze on its seedling/young plants (*Kuijper, Nijhoff & Bakker, 2004*; *Fokkema et al., 2016*). However, short seedlings/young plants may escape from grazing via association with non-preferred plants such as *Artemisia maritima*. Additionally, hares and geese may increase topographic variation *via* trampling (*Van Wijnen, Van Der Wal & Bakker, 1999*; *Elschot et al., 2015*), they may also impact sediment accumulation (measured by clay thickness) by trampling and altering vegetation structure (*Boorman, Garbutt & Barratt, 1998*). Furthermore, because hares and geese play a less important role than tidal water in seed dispersal (*Chang et al., 2005*), it is unlikely that they can increase genotypic richness and diversity by dispersing seeds. Instead, hares and geese may decrease genotypic richness and diversity by promoting clonal spread of *E. atherica* (*Van Der Graaf, Stahl & Bakker, 2005*). Taken together, we also expect that hares and geese may increase ITV through selective grazing and promoting

topographic variation and variation in clay thickness but may decrease ITV by decreasing genotypic richness and diversity.

## MATERIALS AND METHODS

### Study site

A natural successional gradient is present in the back-barrier saltmarsh of the island of Schiermonnikoog (53°30′N, 6°10′E), the Netherlands. Because this island expands eastward, thus the eastern part of the island is younger relative to the western part (*Olff et al., 1997*). The western part of the saltmarsh is enclosed for cattle grazing, while the eastern part of it is grazed by wild small herbivores such as spring staging geese, year-round present hares and rabbits. Hares and geese are the most abundant herbivores (*Van De Koppel et al., 1996*; *Van Der Wal, Kunst & Drent, 1998*; *Van Der Wal et al., 2000a*; *Kuijper & Bakker, 2005*; *Schrama et al., 2015*). Rabbits are very rare. *Kuijper & Bakker (2005)* found that biomass removed by rabbits in 2000 was 6.33% and 0.16% of the total biomass removed by small herbivores at the early and intermediate successional stage, respectively.

We used long-term hare and goose exclosures that were initiated in 1994 (details in *Chen et al., 2019b*). We selected exclosures located at the early and intermediate successional stages, which are approximately 2.5 km apart (Fig. 1A). Vegetation succession has undergone around 30 and 60 years at the early and intermediate successional stage, respectively. Age of the vegetation succession was counted from the year of vegetation establishment to 2016. The year of vegetation establishment was determined by checking aerial photographs (*Olff et al., 1997*). Grazing pressure from hares and geese was ca. 1.5 times higher at the early successional stage than the intermediate successional stage based on year-round dropping count in 2000 and 2016 (see Table S1 for more details; *Kuijper & Bakker, 2005*; *Chen et al., 2019b*; *Chen et al., 2019a*).

Hare and goose exclosures (one per each successional stage) were located in a similar elevation (early stage: $1.42 \pm 0.004$; intermediate stage: $1.44 \pm 0.004$; mean $\pm$ 1se; $N = 24$; m+NAP; Normal Amsterdam Water Level). Exclosures (8 m × 12 m and 6 m × 8 m at the early and intermediate successional stages) were made by chicken mesh (mesh width 25 mm) supported by wooden posts every 3.5 m to exclude hare grazing inside the exclosures. Exclosures were around 1 m in height, ropes were suspended on top of the wooden posts to stop geese flying into the exclosures. At the beginning of the exclosure experiment (1995), *E. atherica* rarely occurred (<2.5%; percent cover) inside and outside the exclosures at these two successional stages. In 2016 (22 years later), vegetation composition differed in the grazed areas and inside the exclosures (Fig. 1; also see Table S2 for species composition for the three most abundant species). Specifically, *Artemisia maritima* was the most dominant plant in the grazed area at these two successional stages, while *E. atherica* was the most dominant plant inside the exclosures.

### Experimental design

To make our results spatially comparable, we sampled within ca. 6 m × 8 m inside and outside the exclosures at both successional stages in June 2017. We chose this size because this is the largest area we can sample inside the exclosures at the intermediate successional

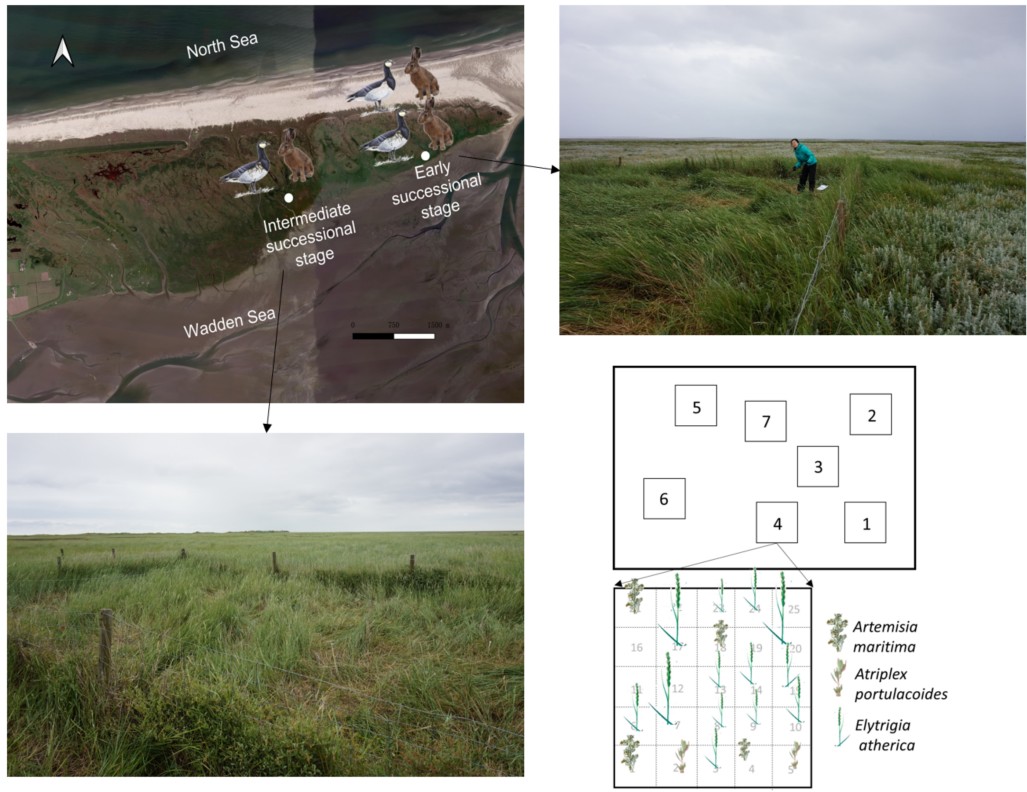

**Figure 1** **Sampling location and scheme.** Location for the 22-year exclosures at the early and intermediate successional stage in the saltmarsh of the island of Schiermonnikoog. Pictures of the exclosures at the early and intermediate successional stages are shown. Sampling plots (1 × 1 m) within an exclosure and sampling scheme for individuals of *Elytrigia atherica* within a 1 × 1 m plot are shown. We followed the same sampling scheme for each plot. *Elytrigia atherica* did not occur in every grid, thus sample size varied for each plot. Sampling plots and scheme were similar outside the exclosures (in the grazed area) at both successional stages. The number of hares and geese indicates the abundance of small herbivores such that the early successional stage had higher grazing pressure (indicated by two hares and two geese) relative to the intermediate successional stage (indicated by one hare and one goose). Note: only the map of the saltmarsh, but not exclosures and sampling plots, is projected according to its actual size.

stage. The distance between the area and the exclosure measured ca. 80 m. We randomly set up 7 plots (1 m × 1 m) inside these two grazed areas and exclosures (Fig. 1). The distance between these plots ranged from 0.5 m to 9 m. We divided each plot into 25 grids (0.2 m × 0.2 m). Within each grid, we collected one individual stem of *E. atherica* (with roots), usually in the middle of the grid (Fig. 1). *Elytrigia atherica* did not occur everywhere, particularly not in the grazed area, thus sample size per plot varied from 9 to 25.

## Trait measurement

We measured traits for individual stems of *E. atherica* in the field. We sampled individual stems without obvious grazing marks. We cut individual stems at the ground level and measured maximum height (cm) from the base to the highest point. We measured the maximum leaf width and length of the first fully grown leaf, usually the third leaf from the top. We also recorded whether individual stems flowered or not. After these measurements,

each stem was stored in a paper bag, sealed, labeled, and then put in a self-sealing plastic bag to reduce water loss in tissues. At the end of each day, samples were brought to the lab, we measured fresh biomass of the individual stems and fully-grown leaves. We also measured their dried biomass (g) after oven-dry (70 °C) to constant mass (ca. 3 days). We calculated specific leaf area ($mm^2$ $mg^{-1}$) as (leaf width $\times$ leaf length)/dried leaf biomass. We calculated leaf dry matter content ($mg$ $g^{-1}$) as dried leaf biomass/fresh leaf biomass. Specific leaf area and leaf dry matter content were not measured in the standard way (*Pérez-Harguindeguy et al., 2016*), and caution should be taken when comparing our data with other studies. Due to that reason, we focus on height, aboveground biomass, and flowering in the main text, we present the results for specific leaf area and leaf dry matter content in the Supplemental Information.

## Genotyping and genotype richness and diversity

We used oven-dried leaf samples (ca. 2 mg per individual stem) for DNA extraction. We first shredded leaf samples into smaller pieces using tissuelyser. We then extracted DNA from each leaf sample using the CTAB method (*Doyle & Doyle, 1987*) and stored DNA samples at −20 °C before PCR. We amplified DNA using PCR with fluorescence-labeled primers. We used five microsatellite markers (ECGA89, WMS6, WMS44, WMS2, and ECGA89) originally designed for the other Poaceae species, *Elymus caninus* (*Sun, Salomon & Bothmer, 1998*) and *Triticum aestivum* (*Röder et al., 1998*). These five markers have been used for genotyping *E. atherica* in this system (*Bockelmann et al., 2003*; *Chen, 2020*). The PCR products from primer ECGA89, WMS6, and WMS44 were pooled together, while the PCR products from WMS2 and ECGA89 were pooled together. Additionally, 1 µL rROX was added in each sample as the internal size standard (Gene ScanTM–350 ROXTM, Applied Biosystem). We visualized the pooled PCR products using the 3730 DNA analyzer and scored the microsatellite peak patterns (height > 100) manually using GeneMapper. In total, we successfully genotyped 579 individual stems of *E. atherica*, but 2 were excluded for further analyses as some trait data measured in the field for these samples were missing.

## Abiotic variables

We measured clay thickness and topographic variation for each plot (3 replicates). We measured clay thickness using a 2 cm Ø soil corer with tick marks as a proxy for soil total nitrogen (*Olff et al., 1997*). Previous studies show that clay thickness is strongly positively correlated with soil total nitrogen in this system (*i.e.,* total nitrogen (g) = 186 + 43 clay thickness (cm); $r = 0.9$, $p < 0.0001$; (*Olff et al., 1997*)). Also, clay thickness is used as a proxy of soil fertility (*e.g.*, *Schrama et al., 2017*). We measured topography (elevation) using Trimble R8 (precision for elevation ca. 1 cm).

# DATA ANALYSIS

## Calculating trait means, ITV, topographic variation, and variation in clay thickness

Although our focus is on ITV, to compare whether the effects of small herbivores on ITV are stronger than trait means, we also looked at trait means. We calculated means for each

trait by averaging trait values over 9 -25 individual stems in each plot. We calculated ITV in each trait each plot as the standard deviation /mean. Topographic variation and variation in clay thickness were calculated as standard deviation /mean (over 3 samples) of elevation and clay thickness, respectively.

## Calculating genotypic richness, genotypic diversity, and genetic differentiation

To calculate genotype richness and diversity per plot, we first calculated pairwise genetic distance using Dice dissimilarity from the R package ade4 (*Dray & Dufour, 2015*) based on the presence/absence matrix of 42 allele bands from those five markers. We then assigned genotypes based on dice dissimilarity, using the function "assignClones" from the R package polysat (*Lindsay, Clark & Clark, 2018*). We calculated genotypic richness as the number of unique genotypes detected divided by the number of individual stems genotyped for each plot. Genotypic diversity—taking into account the abundance of different genotypes—was calculated using the function "genotypeDiversity" with the index of "Shannon" from the package polysat (*Lindsay, Clark & Clark, 2018*). Small herbivores may select for some particular genotypes that are more resistant or tolerant to herbivore grazing (*Kotanen & Bergelson, 2000*), which may also impact traits. Therefore, we explored genetic differentiation using principal coordinates analysis (PCoA) from the R package ade4 (*Dray & Dufour, 2015*). We looked at genetic differentiation at the treatment level, that is, grazed and ungrazed across plots at each successional stage, result can be found in Fig. S1.

## Effects of small herbivores on trait means and ITV

We used analysis of variance, function "lm", to look at the effects of small herbivores on means and ITV of each trait at the early and intermediate successional stage, separately. We checked model assumptions by visually inspecting residual plots for homogeneity and quantile–quantile plots for normality. In all cases, models fitted the data reasonably well.

## Direct and indirect effects of small herbivores on ITV

We estimated the direct, indirect effects through altering genotypic richness and diversity, indirect effects through altering abiotic variables, and total effects (sum of direct and indirect effects) of small herbivores on ITV. We estimated these effects for each trait at each successional stage based on the standardized path coefficients using structural equation models from the R package "lavaan" (*Rosseel, 2012*). Note that leaf dry matter content was not evaluated in structural equation models due to limited sample size (see online supporting text). Grazing (0: ungrazed, 1: grazed), genotypic richness, genotypic diversity, topographic variation, and variation in clay thickness were included in the models. Note that genotypic diversity was not included for ITV at the early successional stage, as it was significantly correlated with genotypic richness (variance inflation factor > 4). See online supporting text for an example of a structural equation model for ITV in height at the intermediate successional stage and calculation of the direct and indirect effects. Effects are significant when $p \leq 0.05$, while marginally significant when $p > 0.05$ and $p < 0.1$. Data analysis was performed in R3.5.3 (*R Core Team, 2020*).
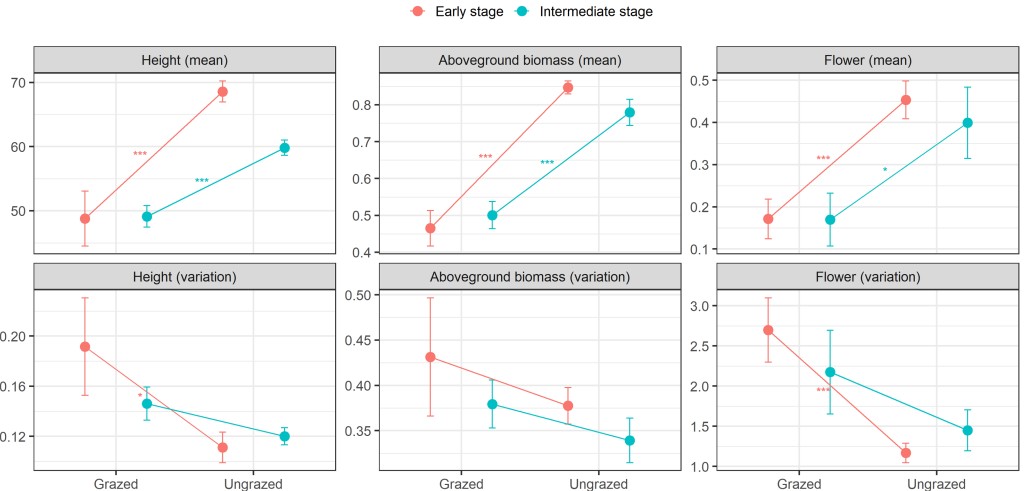

**Figure 2** **Means and variations in functional traits of individual stems of *Elytrigia atherica* in the grazed and ungrazed plots at the early and intermediate stages.** Traits include height (cm), aboveground biomass (g), flower frequency. Results for specific leaf area ($cm^2\ g^{-1}$) and leaf dry matter content ($g\ g^{-1}$) are not shown because small herbivores generally did not impact the means and variations in these two traits at both successional stages. Dots are the means, error bars show 1 se. Asterisks indicate significant levels: * $P < 0.1$; ** $P < 0.05$; *** $P < 0.001$. See Table S3 for test statistics.

## RESULTS

### Effects of small herbivores on trait means and ITV

At the early successional stage, small herbivores significantly decreased means in height, aboveground biomass, and flowering of *E. atherica*. Also, small herbivores significantly increased ITV in flowering and marginally increased ITV in height, but had no effects on ITV in aboveground biomass. At the intermediate successional stage, small herbivores significantly decreased means in height, aboveground biomass, and flowering. However, small herbivores had no effects on ITV in these traits (Fig. 2; see Table S3 for test statistics). Small herbivores did not impact mean and ITV in specific leaf area and leaf dry matter content at both successional stages except that small herbivores decreased mean in specific leaf area at the intermediate successional stage (Table S3).

### Direct and indirect effects of small herbivores on ITV

The structural equation model reveals that overall (summing up the direct and indirect effects), small herbivores tended to promote ITV in height and flowering, but not in other traits at the early successional stage. While, small herbivores tended to promote ITV in height, this did not occur in other traits at the intermediate stage. Moreover, these positive overall effects of small herbivores on ITV in height and flowering were mainly attributable to their direct effects. Although the overall effects of small herbivores on ITV in aboveground biomass and specific leaf area were not significant, small herbivores also directly significantly impacted ITV in these two traits at the early successional stage (Fig. 3; Fig. S2). See Table 1 for the standardized path coefficient estimated from SEM for the direct, indirect, and total effects of small herbivores on ITV.

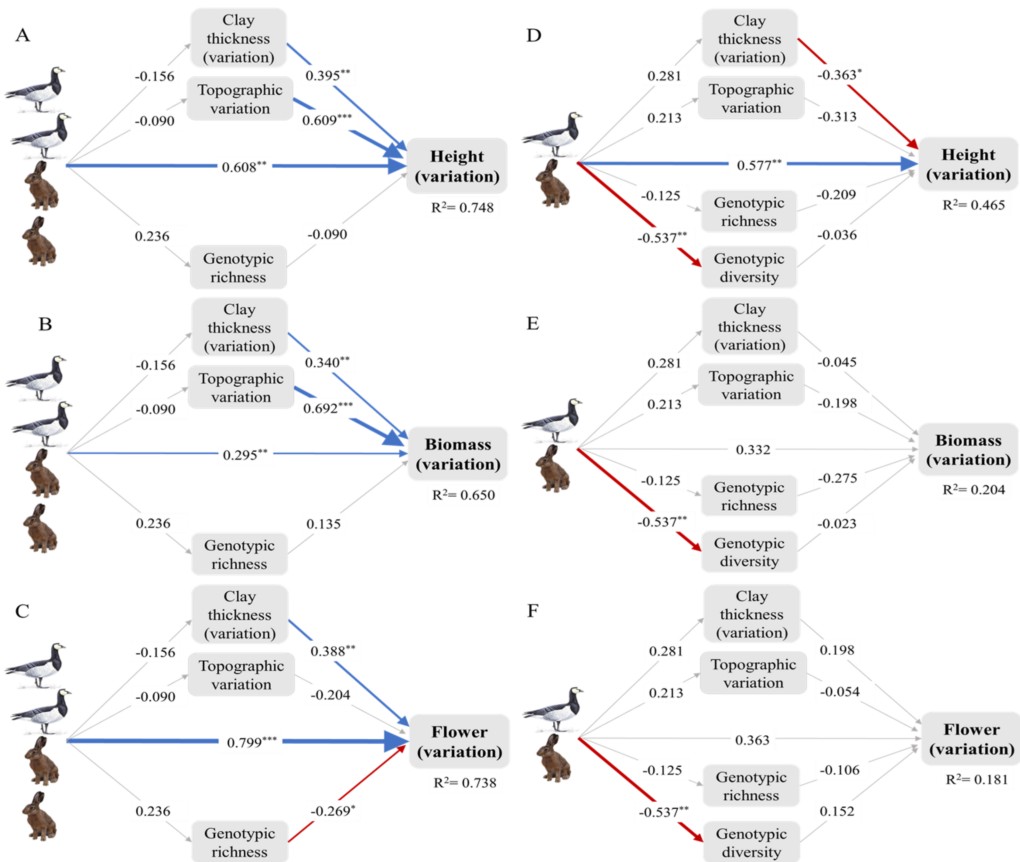

**Figure 3 Intraspecific trait variation (ITV) of the dominant plant *Elytrigia atherica* and the direct and indirect effects of small herbivores on ITV in local communities at the early (A-C) and intermediate stage (D-F).** The direct effects, indirect effects through genotypes, indirect effects through abiotic variations, and total effects of small herbivores on ITV in each trait are summarized in Table 1. Model fit the data well (for all models at the early successional stage: $\chi^2 = 4.409$, $df = 3$, $N = 14$, $p > 0.05$; for all models at the intermediate successional stage: $\chi^2 = 6.559$, df = 6, $N = 14$, $p > 0.05$). Variance explained for clay thickness (variation), topographic variation, and genotypic richness for models at the early successional stage are 0.024, 0.008, and 0.056, respectively. Variance explained for clay thickness (variation), topographic variation, genotypic richness, and genotype diversity for models at the intermediate successional stage are 0.079, 0.045, 0.016, and 0.289, respectively. Number of hares and geese indicate the abundance of small herbivores such that the early successional stage had higher grazing pressure (indicated by two hares and two geese) relative to the intermediate stage (indicated by one hare and one goose). Boxes are measured variables. Arrows denote unidirectional relationships among variables. Blue arrows are significant positive relationships, red arrows are significant negative relationships, and grey arrows show non-significant relationships. The width of the arrows indicates the strength of the pathways. The values on the arrows denote standardized path coefficients. Asterisks indicate significant paths: * $p < 0.1$; ** $p < 0.05$; *** $p < 0.001$. See Fig. S2 for the direct and indirect effects of small herbivores on ITV in specific leaf area at the both successional stages. Note that leaf dry matter content was not evaluated in structural equation models due to limited sample size (see online supporting text).

At the early successional stage, small herbivores did not have significant effects on genotypic richness and diversity, but genotypic richness decreased ITV in flowering. At the intermediate successional stage, small herbivores did not have significant effects on genotypic richness, but significantly reduced genotypic diversity. However, genotypic

**Table 1 Direct and indirect effects of hares and geese on intraspecific trait variation (ITV) of *Elytrigia atherica* at the early and intermediate successional stages.** See online supporting text for an example of how these data were calculated and how significance was determined using SEM.

| Successional stages | Traits | Direct effects | Indirect effects through genotypes | Indirect effects through abiotic variations | Total effects |
|---|---|---|---|---|---|
| Early | Height | 0.608*** | −0.021 | −0.117 | 0.47** |
| | Biomass | 0.295** | 0.032 | −0.116 | 0.211 |
| | Flowering | 0.799*** | −0.063 | −0.042 | 0.694*** |
| Intermediate | Height | 0.577** | 0.045 | −0.169 | 0.454* |
| | Biomass | 0.332 | 0.022 | −0.055 | 0.299 |
| | Flowering | 0.363 | −0.068 | 0.044 | 0.339 |

**Notes.**

Asterisks indicate significant effects.

*$p < 0.1$.

**$p < 0.05$.

***$p < 0.001$.

richness and diversity did not impact ITV in any traits at the intermediate successional stages (Fig. 3; Fig. S2).

Small herbivores did not impact topographic variation and variation in clay thickness at both successional stages. However, these variations significantly increased ITV in all traits at the early successional stage except that the effects of topographic variation on ITV in flowering were not significant. The effects of these variations on ITV were less apparent at the intermediate successional stage except that variation in clay thickness marginally decreased ITV in height (Fig. 3; Fig. S2).

## DISCUSSION

In this study, using long-term exclosures, in combination with genetic analyses, we explored ITV of a dominant clonal plant (*Elytrigia atherica*) and the direct and indirect effects of small herbivores on ITV. We found that, at the early successional stage, small herbivores significantly ITV in height and flowering of this plant. At the intermediate successional stage, small herbivores marginally promoted ITV in height. Moreover, small herbivores promoted ITV mainly through direct effects (*e.g.*, selective grazing) but not through altering genotypic richness/diversity or topographic variation and variation in clay thickness.

### Effects of small herbivores on trait means and ITV

Our results suggest that small herbivores could impact both means and variations in key functional traits of *E. atherica*, but their effects differed in different traits. At the early successional stage, we found that small herbivores significantly decreased means in three traits measured (height, aboveground biomass, and flowering) and they significantly promoted ITV in two traits (height and flowering). At the intermediate successional stage, small herbivores significantly decreased means in four traits (height, aboveground biomass, flowering, and specific leaf area) and they marginally promoted ITV in one trait (height; Fig. 2; Fig. S2). This suggests that trait means and variations may be driven by different processes. Moreover, we only observed ITV in height and flowering may be because these

two traits have higher plasticity than other traits, and traits with a higher level of plasticity show higher ITV (*Givnish, 2002*). Furthermore, small herbivores had similar effects on trait means at both successional stages, but had stronger impacts on ITV at the early successional stage. This may be because the effects of small herbivores on trait means and variations were mediated by community composition in the field. At the early successional stage, the preferred plant, *F. rubra*, was more abundant in the grazed area (ca. 3 times relative to that of the intermediate successional stage), thus small herbivores (although more abundant) may graze more on *F. rubra* rather than *E. atherica*. At the intermediate successional stage, the abundance of *F. rubra* deceased, while the abundance of *E. atherica* increased in the grazed area (ca. 5 times relative to that of the early successional stage), thus small herbivores (less abundant) may also substantially grazed on less preferred *E. atherica*. Therefore, overall, the effects of small herbivores on trait means were similar at these two successional stages. Meanwhile, non-preferred species (*A. maritima*) was more abundant in the grazed area at the early successional stage (ca. 2 times relative to that of the intermediate successional stage; see Table S2 for detailed percent cover of these species). It may facilitate young *E. atherica* escape from herbivore grazing. Therefore, ITV was more apparent at the early successional stage than the intermediate one. Previous studies in this system have shown that hares and geese are the important drivers for plant community composition and structure along this successional gradient (*Olff et al., 1997*; *Kuijper & Bakker, 2005*; *Chen et al., 2019b*). Here, extending the current knowledge, we show that small herbivores may also impact trait means and variations, which in turn could impact community composition and structure (*Whitlock, Grime & Burke, 2010*).

## Direct and indirect effects of small herbivores on ITV

As we hypothesized, the direct effects of small herbivores through selective grazing increased ITV. This is not only true for height and flowering but also for aboveground biomass and specific leaf area, despite the overall effects of small herbivores on these two traits were not significant. As explained in the previous paragraph, the direct effects of small herbivores may be mediated by community composition especially the proportion of preferred and non-preferred plants. *Herz et al. (2017)* found that local neighborhood diversity can explain a large amount of ITV in German meadows and pastures, possibly through increased plant-plant interactions (*e.g.*, competition). In this study, plant diversity was higher in the grazed than the ungrazed plots at both successional stages after 22-year grazing (Table S2). Thus, higher plant diversity may also contribute to increased ITV in height under grazing at both successional stages. Future studies/experiments looking at the effects of herbivores on ITV in plant communities with the same composition are needed to fully separate the effects of selective grazing from the effects of neighboring plants.

We found that small herbivores only decreased genotypic diversity at the intermediate successional stage. A previous study in the west part of this saltmarsh found that cattle grazing does not impact genotype richness and diversity of this plant, but cattle grazing alters its morphological traits in the field such as decreasing height and leaf width (*Veeneklaas et al., 2011*). Here, we found that genotypic richness and diversity generally had no significant effects on ITV. Microsatellite markers are commonly used in studying plant genetics (*Jarne*

& *Lagoda, 1996*; *Vieira et al., 2016*), but because they are selectively neutral, they may not control gene expression for the traits measured here. More generally speaking, the genetic basis of most traits in many wild plants is unknown. Thus, it is difficult to know in advance which genes or genomic regions to consider, which further hinders inferring a causal relationship between genetic variation and phenotypic variation (*Merilä & Hendry, 2014*). A low correlation between variation in morphological traits and DNA markers was also reported before (*Kolliker et al., 1998*). Additionally, we found genetic differentiation of *E. atherica* in the grazed and ungrazed areas at the early successional stage, but not at the intermediate successional stage (Fig. S1). However, more data (presumably from more exclosures) are needed to consolidate this result. Thus, the observed trait changes (at both successional stages) may not be associated with genetic changes in *E. atherica*.

In contrast, we found no effects of small herbivores on topographic variation and variation in clay thickness within plots. These abiotic variations may be primarily induced by flooding and inundation in this saltmarsh, which may overrule the effects of small herbivores. However, variations in these two abiotic variables had substantial positive impacts on ITV particularly at the early successional stage. Abiotic environment may be more heterogenous at the early successional stage because this stage has more frequent sediment deposition than the intermediate successional stage (*Schrama, Berg & Olff, 2012*). A previous study conducted in the western part of this saltmarsh also found that topographic variation at the small spatial scales (0.1 m$^2$ and 10 m$^2$) is positively correlated with species richness and cattle grazing additionally increased species richness (*Ruifrok et al., 2014*). Thus, abiotic variations, even at very small spatial scales, may play an important role in promoting ITV and altering other plant community properties.

### Long-term exclosures

Although we used 22-year old herbivore exclosures, our data cannot answer whether duration of herbivore grazing impacts ITV, as we only collected data for one year. To our knowledge, no studies have compared the effects of short- and long-term grazing on ITV. *Didiano et al. (2014)* found that tolerance to rabbit grazing decreased as the age of the exclosures increased in *F. rubra*, the most abundant plant in Silwood Park, England. Therefore, grazing duration may also impact ITV.

## CONCLUSION

Our results suggest that small herbivores may not only impact means but also ITV in some key functional traits of a dominant plant (*E.atherica*). Additionally, small herbivores impacted ITV mainly through selective grazing but not through altering genotypic richness/diversity and abiotic variations. However, topographic variation and variation in clay thickness may contribute to ITV. Small herbivore populations are changing rapidly due to human actions. For instance, populations of European brown hares have declined dramatically due to land-use changes (*Smith, Jennings & Harris, 2005*), while populations of geese are rapidly increasing globally (*Menu, Gauthier & Reed, 2002*). These changes in small herbivore populations could thus impact their effects on ITV in plant species, which may have consequences for plants to respond to increasing environmental changes.

## ACKNOWLEDGEMENTS

We thank Iris Bontekoe, Erica Zuidersma for helping collect samples in the field. We thank Marco van der Velde, Jan Veldsink, and Yvonne Verkuil for their help with genotyping in the lab. We thank JF Scheepens for his constructive comments on the earlier versions of this manuscript. We thank Juan Alberti and Oliver Carroll for their constructive suggestions for improving the structure and readability of this manuscript. We thank three anonymous reviewers and the associate editor for their constructive comments and suggestions. We thank Natuurmonumenten for offering us the opportunity to work in the saltmarsh of the island of Schiermonnikoog.

### Funding

QC is funded by CSC (China Scholarship Council). The funders had no role in study design, data collection and analysis, decision to publish, or preparation of the manuscript.

### Competing Interests

The authors declare there are no competing interests.

### Author Contributions

- Qingqing Chen conceived and designed the experiments, performed the experiments, analyzed the data, prepared figures and tables, authored and reviewed drafts, and approved the final draft.
- Christian Smit and Han Olff contributed to the idea of this project, reviewed drafts, and approved the final draft.
- Ido Pen facilitated data analyses, reviewed drafts, and approved the final draft.

### Data Availability

The data is available at Figshare: Chen, Q., Smit, C., Ido, P., Olff, H., (2021): Data from Small herbivores and abiotic heterogeneity promote trait variation of a saltmarsh plant in local communities. figshare. Dataset. https://doi.org/10.6084/m9.figshare.14740386.v1.

### Supplemental Information

Supplemental information for this article can be found online at http://dx.doi.org/10.7717/peerj.12633#supplemental-information.

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
