# Peer review of "Small herbivores and abiotic heterogeneity promote trait variation of a saltmarsh plant in local communities"

_PeerJ, doi:10.7717/peerj.12633_

## Round 0.1 · original submission · Major Revisions

Three recognized experts have assessed your manuscript and identified a number of issues that the manuscript revision should resolve. In order to be accepted, the manuscript needs a comprehensive and far-reaching revision. This is particularly relevant for the presentation of the results and the problem identified by reviewer 3 that the methodology may not be adequate to obtain robust results. Furthermore, your conclusions seem not to be sufficiently supported by data presented. Finally, the discussion needs to describe in more detail the potential mechanistic response with respect to the direct and indirect effects found.
Please consider the proposed changes as mandatory.

·

Basic reporting

The manuscript authored by Chen et al. entitled “Small herbivores and abiotic heterogeneity promote trait variation of a saltmarsh plant in local communities” uses a 22 years experimental approach to test if the presence of hares and geese have impact on the intraspecific trait variation (ITV) of Elytrigia atherica in two successional stages. They analyze height, aboveground biomass, SLA, leaf dry material content and the presence of flowers to study the ITV. Also they study genetic diversity of the mainly-clonal plant species.

The effect of this herbivorous species had a direct effect on several traits, having more marked effects at early successional stages (where the herbivory is also higher), some of the abiotic variations and genotypes also affects these traits, but are not affected by the herbivorous so no indirect effect was detected in this study.

The language used by the authors is clear, unambiguous and professional. The approximation to the questions provided in this article follow a logical order with relevant references. They introduce first the intraspecific trait variation, then explain that herbivorous can affect this variation and then they focus on the small herbivorous effects on this ITV to finally focus on hares and geese to propose the hypothesis that these small herbivorous can increase the ITV of E. atherica. I found Figure 2 very illustrative and inspiring of the results obtained, although a little repetitive. For every trait, a different diagram is represented although the effect of the herbivorous on the indirect effects is always the same. I would prefer to have only two diagrams, one per successional stage (as long as the authors can find a solution to allow the correct interpretation of the results, avoiding unnecessary overlaps between effects). Also the resolution of the figure is quite low, maybe as a result of uploading constraints.
The raw data only include the variation of traits, but not the values of this traits.

Experimental design

The experimental design is clear and well builded, not including more constraints than necessaries. Although the design has been used previously in other articles (see Chen et al. 2019) not overlap of the results were shown. The research question is also clear, and the experimental design is properly designed to answer it. The methods are properly explained and it is easy to replicate them.

Validity of the findings

The results here provided open the gate to study the effect of small herbivorus mammals (or other type of herbivorous) in the ITV of plants, with the evolutive consequences this could have. The results are also robust since 22 year dataset were used to this article. The conclusions are linked to the original question.

Additional comments

General notes and specific comments

There is a problem with the references and the bibliography section, with several studies lacking from the text or the bibliography. The proportion of new references (less than 5 years) almost reach the 50% of all the references used for this article.

L45: He and Silliman, 2016 is not cited in the bibliography

L47: Louault et al., 2005 instead of “Louault, Pillar, Aufrère, Garnier, & Soussana, 2005”

L50: I do not understand the concept of “non-borrowing invertebrates”. Also hares and geese are not invertebrates. Finally, I do not find accurate the use of “small” referring to animals between 1 and 10 kg weight. In the published articles I have consulted (Olff and Ritchie, 1998; Olofsson et al., 2004; Firn et al., 2017, 2019; Jessen et al., 2020) small herbivorous is used to define less than 1 kg weight herbivorous (e.g. voles) and others like rabbits are defined as intermediate-sized herbivorous, but in Chen et al. 2019 hares and geese are referred as small vertebrate herbivorous. I find confusing the “small herbivorous” concept and I think that adding “vertebrate”, as it was included in the abstract, it would be enough to make it clear and avoid readers to think, e.g., on invertebrate herbivorous species.

L65: Rico & Wagner, 2016 is not cited in the bibliography

L77: “Olff et al., 1997” instead of “Olff, Leeuw, Bakker, Platerink and van Wijnen, 1997”

L83: “Kuijper et al., 2004” and “Fokkema et al., 2016” are not cited in the bibliography

L93: “Boorman et al., 1998” instead of “Boorman, & Garbutt, & Barratt, 1998” and also is not cited in the bibliography

L95: “Olff et al., 1997” instead of “Olff, Leeuw, Bakker, Platerink and van Wijnen, 1997”

L102: “Chen et al., 2019” instead of “Chen et al. 2019”

L117: “Van De Koppel et al. 1996” is not cited in the bibliography, and also “Van De Koppel et al., 1996” instead of “Van De Koppel et al. 1996”

L118: “Van Der Wal et al. 1998” and “Van Der Wal et al. 2000a” are not cited in the bibliography

L118: “Schrama et al., 2015” instead “Schrama et al. 2015”

L122: “Chen et al., 2019” instead of “Chen et al. 2019”

L173: “Doyle & Doyle, 1987” is not cited in the bibliography

L174: “Bockelmann et al. (2003)” instead “Bockelmann et. al (2003)”

L176: “Sun et al., 1998” instead “Sun, Salomon, & Bothmer, 1998”

L177: “Bockelmann et al. (2003)” instead “Bockelmann et. al (2003)”

L189: “Lindsay et al., 2018” instead of “Lindsay, Clark, & Clark, 2018” and also is not cited in the bibliography

L193: “Lindsay et al., 2018” instead of “Lindsay, Clark, & Clark, 2018” and also is not cited in the bibliography

L197: “Olff et al., 1997” instead “Olff et al. 1997”

L277-278: “Schrama et al., 2012” instead of “Schrama, Berg, & Olff, 2012” and also is not cited in the bibliography.

L287: “Didiano et al. (2014)” is not cited in the bibliography

L301: “Smith et al., 2005” instead of “Smith et al. 2005” and also is not cited in the bibliography

L301-L302: “Menu et al., 2002” instead of “Menu et al. 2002” and also is not cited in the bibliography

L368: “Firn, J., J. M. McGree, E. Harvey, H. Flores-Moreno, M. Schütz, Y. M. Buckley, E. T. Borer, et al. 2020. Author Correction: Leaf nutrients, not specific leaf area, are consistent indicators of elevated nutrient inputs (Nature Ecology & Evolution, (2019), 3, 3, (400-406), 10.1038/s41559-018-0790-1). Nature Ecology and Evolution 4: 886–891.” is not cited in the text

·

Basic reporting

The manuscript has all the basic standards for publishing a quality article. English seems to be at a high level, but I'm not a native speaker, so I leave this to the editor.
The formatting of the references seems to be fine, but the citations in the manuscript are not standardized. The order of citations is not unified, eg Line 36-37 seems to be by year, but elsewhere it is alphabetical. Furthermore, there is no uniform use of the symbol “&” versus “and” in the citation.
The structure of the manuscript is fine, however, the fact that the manuscript is based on some previous studies is a bit confusing, and the reader is not sure when reading the methodology if he is no longer reading the results.
The structure of the subchapters in the Methodology is not clear - Some chapters are in italics and something is bold. In addition, some texts of subchapters are just after the title, and elsewhere otherwise. Arrange according to the rules of the journal.
The conclusion is in italics, shouldn't it be bold like other chapters?
A few minor errors:
Line 145 - the gap between 1 and m
Line 197 - Olff et al. 1997 - complete the comma
Line 278 - Herz et al., - delete comma

Experimental design

The design of the experiment seems to be very well managed, and even the data processing seems to be fine.

Validity of the findings

As I have already written in the previous sections, the manuscript contains very well processed long-term data and is very well written. However, that's all, the interpretation of the results lags behind its possibilities. The authors beautifully explain the consequences of the obtained data, but the cause of this condition is no longer addressed anywhere. Why is selective grazing? Why do small herbivores choose specific genotypes of this plant? The literature (which is missing here, eg: Bailey et al. 1996 Journal of Range Management; Wallis De Vries et al. 1999 Oecologia; Evju et al. 2009 Oecologia; Matches (1992 Journal of Production Agriculture, et al.) is full of information about the reason for selection by different herbivores.
Herbivores: for generalists, we have two main selection pressures for this kind of selection, (1) mechanical defense of the plant (high density of trichomes, etc.); and or (2) chemical defense (increased amounts of tannins, etc. - young versus old leaves, etc.). Within one species of plant, a high variability of these traits is known (influence of genotype and/or the age of the plant). Although you did not get any data that would support it in any way, it would be very appropriate to mention it in the discussion and in the introduction. Plants can thus significantly affect the size of the herbivore population (eg Saska et al. 2021 Journal of Pest Science).

Reviewer 3 ·

Basic reporting

Results and hypothesis need to be clarified (see General comments for the author)

Experimental design

I have some concerns regarding the Method section (see General comments for the author)

Validity of the findings

Conclusions need improvements and some methods used or results do not allow to draw certain conclusions that the authors present (see General comments for the author)

Additional comments

In this manuscript, the authors investigated whether small vertebrate herbivores affect the variability of phenotypic traits of the clonal grass Elytrigia atherica, either by direct grazing or indirectly by altering genotypic diversity or soil conditions due to its interaction with the environment. To do so, the authors used exclosures installed in plant communities of a saltmarsh in the Netherlands, in order to compare the intra-specific variability and patterns of neutral molecular markers inside and outside (open to herbivore’s grazing) the exposures. SEM models showed significant positive direct and indirect (through changes in soil abiotic conditions) relationships of herbivore pressure with plant traits variation (ITV) in early successional stages, while genetic diversity was reduced by herbivores in intermediate stages, without any significant change on ITV.

The question on how herbivory can affect the phenotypic variation and promote a change in the genetic diversity of a clonal species is interesting. The use of Path Analysis (i.e., SEM) is also appropriate to investigate indirect effects from herbivory activities. However, the methodology used seems not always the most appropriate for testing their hypothesis or at least does not allow to draw certain conclusions that the authors present. Moreover, in my opinion, some parts of the manuscript lack clarity due to inaccuracies in the writing and the way tables and figures are presented and the results and discussion need further development.

Major comments
1.- I have some concerns regarding the Method section. Firstly, the authors selected only one exclosure per successional stage so no replicates exist in this matter. This limits the possibility to present general patterns, so the conclusions must be presented considering this limitation. Second, you did used different sizes of exclosures between early (96 m2) and intermediate stages (48m2), which may be accounted for while drawing the conclusions. Third, according to L160-166, some of the leaf functional traits seem not correctly measured, as the authors did not used standard methods or only some plots were measured. This needs further explanation since you then have a biased measure and an inconsistent data collection that may affect your results.
In addition, despite I am not an expert in molecular analysis, I would assume that the genetic diversity inferences from neutral molecular markers may be affected by changes in sexual/clonal reproduction (mentioned in L85-86) rather than by herbivory itself (as examples for clonal species, see Holderegger et al. 1998; Bruun et al. 2007). This means first, that using neutral markers it would be unlikely to find that ITV is driven by the selection of these markers because they cannot control the gene expression of these traits, and second, that differential grazing selection considering plant size may be questionable, as the authors mentioned in L59-60 (or a reference is needed to corroborate this). Otherwise, plants clonal diversity might be affected by grazing, especially at early stages, but you only included one control, right?
2.- The results need a more appropriate and detailed description. Even, according to the results presented in Fig. 2 and Table 1, they are incomplete. For instance, the authors only mention height and flowering appearance as the traits affected by direct effects of herbivory in early successional stages. However, I see other significant positive effects in this case such as biomass and SLA. Each of these trait changes needs to be presented and discussed considering your predictions. Also, I would recommend to include the results of the mean values in the main text and discuss accordingly, as you commented that “Small herbivores may have more pronounced effects on plants: they not only impact trait means as studies have shown but also ITV”. This question is interesting but not discussed at all in the text (see for instance studies evaluating this but in another framework: Herrera 2017, March-Salas et al. 2021).
3.- Considering the previous points and other issues, I believe that the discussion should be toned down when considering global changes (e.g., in L249) and better defined to describe the exact conclusions that can be drawn from the analysis carried out and results. In your study, intra-specific variation could be also explained by phenotypic plasticity or genetic differentiation and this is little discussed. I also strongly recommend to compare more in deep the relevance of changes in means versus changes in trait variance (as examples, see previously mentioned studies), and discuss your findings with other closely related studies (e.g., Veneklaas et al. 2011) or others (see references).

Minor comments:
L11-12: Before mentioning that little is known about this topic, I would introduce first why this is a relevant question for plant ecologists. For instance, at this point, I cannot connect the effect of small herbivores in ITV with global changes.
L12: Change “response” by “respond”.
L16: You mean grazing outside the exclosure, right?
L17: You “measure” functional traits (also see L102).
L20: The meaning of “elevation” needs to be clarified, and please use a more specific name.
L27: I would remove “have more pronounced effects on plants: they”.
L39: Please specify in some point the differential relevance between genotypic diversity and genotypic richness in the context of your study.
L50: They are vertebrates and not “invertebrates, right?
L55-56: If this is not a question of your research, please remove.
L69: With environmental heterogeneity you refer to soil conditions (nutrients, texture, compaction…), correct? And what do you refer in L71 with microenvironmental heterogeneity?
L72-73: I would remove this sentence because you already write in different parts of the text that your question is underexplored.
L80: Use “preferring” and not “preferred”.
L82-83: This seems contradictory with what you said in L59-60.
L84: “hiding” doesn’t seems the correct term.
L85: Following grammar rules, after the point, the complete genus name should be written see also line 149).
L90: I would use “spread” instead of “growth”.
L96-98: The hypotheses are barely exposed or not clearly differentiated. I also miss here the connection of your results with changes in trait means, since this is one of the main conclusions that the authors exposed.
L112-113: I believe that this sentence is not correctly exposed. Please, rephrase.
L114-117: Despite the mentioned abundance, you cannot disentangle between the effects of hares and geese from the effects of other existing small herbivores such as rabbits, so I would keep it more open to the effect of small herbivores. Please, revise this along the entire text, especially in the results.
L121: To my knowledge, this is not an experimental system and mostly and study system. Also, you did not describe your measurements here.
L132: Say that this is Means ± SE, if this is the case.
L136: Mention when this experiment started.
L152: It was strange to me to read that you only measured in “sunny days”.
L153: I cannot follow this sentence. So please, rephrase.
L160: remove “within hours”. Also, 70 ºC seems high temperature for this purpose. I would do that at 36-42ºC.
L163: In addition to what I commented above, I could imagine than using dried samples instead of fresh material or samples with ethanol would affect the result of the DNA-sequencing.
L203: To facilitate reader’s comprehension, I would try to connect statistical analyses with the hypothesis stated in the introduction.
L221 and others: As commented before, I would use herbivory instead of hares and geese.
L221-222: Include here the p-value and also the statistic.
L222-223: I would be more neutral: “At intermediate stage, herbivores tended to affect ITV in height in grazed areas… And direct effects should be described more in deep in the next sentence.
L235: This was not the case in the effect of elevation in the variation of flower appearance.
L259: I do not think that the effect of microbial community can be related to your data/results.
L281-282: Which results indicate this response? I don’t think you can predict this with your design.
L293: You mainly summarize your results again instead of exposing the concluding remarks.
L295: Please, remove “substantial”.
L299: Maybe mention human actions is more appropriate that to relate this to global changes.
Fig. 1- I believe that the figure can be substantially improved. It could be presented as an infographic and not as separate panels. Also, as for panel D, is this an example? I would assume that this structure differs among plots. Also, what do the animals and the animals number represent? And the bar in panel B is the scale. The blue colours in D-F reduce the visualization too. Include this detail in the figure text. Panel F has the area size but not the Panel E. By the way, why not present the areas in early stages?
Fig. 2- I am missing the R2 values. Include the letter “H” in the last panel. Again, what do the animals and the animals number represent?
Fig. S2- Show significance and use letters. Maybe this figure can be moved to the main text. Also, why did you describe the results here? Cpuld them be moved to the main text?
Table S3- How do you explain that you find the same results in all the traits within each successional stage?

References
Bruun, H. H., Scheepens, J. F., & Tyler, T. (2007). An allozyme study of sexual and vegetative regeneration in Hieracium pilosella. Botany 85(1), 10-15.

Henn et al. (2018). Intraspecific trait variation and phenotypic plasticity mediate alpine plant species response to climate change. Frontiers in Plant Science 9, 1548.

Herrera (2017). The ecology of subindividual variability in plants: patterns, processes, and prospects.
Web Ecology 17(2), 51-64.

Holderegger, R., Stehlik, I., & Schneller, J. J. (1998). Estimation of the relative importance of sexual and vegetative reproduction in the clonal woodland herb Anemone nemorosa. Oecologia, 117(1-2), 105-107.

Laforest-Lapointe, Martínez-Vilalta & Retana (2014). Intraspecific variability in functional traits matters: case study of Scots pine. Oecologia 175, 1337–1348.

March-Salas, Fandos & Fitze (2021). Effects of intrinsic environmental predictability on intra-individual and intra-population variability of plant reproductive traits and eco-evolutionary consequences. Annals of Botany 127(4), 413–423.

Santana et al. (2021). Edge effects in forest patches surrounded by native grassland are also dependent on patch size and shape. Forest Ecology and Management 482, 118842.

Veneklaas, R. M., Bockelmann, A. C., Reusch, T. B., & Bakker, J. P. (2011). Effect of grazing and mowing on the clonal structure of Elytrigia atherica: a long-term study of abandoned and managed sites. Preslia 83(3), 455-470.

---

## Round 0.2 · Minor Revisions

We are almost there. Please consider the remarks and proposals of reviewer 3 in a (hopefully) final revision of your manuscript.

·

Basic reporting

The authors precisely solved everything that was commented on in the first version. I have no further comments at this time.

Experimental design

.

Validity of the findings

.

Reviewer 3 ·

Basic reporting

The revised manuscript includes substantial improvements compared with the previous version. The literature references are better presented thanks to the comments done by reviewer 1 and 2 and authors use professional English and structure. However, the hypothesis needs to be more clearly presented and considering methodological limitations, and other questions need to be attained (see detailed comments below)

Experimental design

Research question is relevant, but, as reported below, I consider that the research question and methods used still need some clarifications and a proper justification

Validity of the findings

I find several similarities with the study of Veeneklaas et al (2011) but authors used a new approach for analysing potential direct and indirect effects on this plant species and its structure. Some statistical questions need to be responded and showed in more detail to assure adequacy in the models used. Conclusions improved in the new revised version but I consider that they still need some clarifications

Additional comments

First of all, I salute the effort of the authors for implementing considerable changes in the text and to clarify my comments/questions, and of the other reviewers too. From these changes on the revised manuscript, I have some comments:
-I still consider that the hypotheses need to be more clearly formulated. The background need to be exposed in advance and not after each hypothesis, because this makes the questions difficult to be followed and masked. I strongly recommend that, in the latter part of the introduction authors should clearly present the predictions from the theoretical framework and according to your design and analysis instead of including them in the middle of the text.
-In your responses and in the main text (line 202-203), you mentioned that other studies reported that early stages are subjected to higher grazing pressure than intermediate stage, and that this was the reason of your selected areas, but any reference was added here. Also, I recommend avoiding over-interpretations as “much higher” if these affirmations are not corroborated by numbers. Similarly, this was also the case when you use “strongly impact” (e.g. line 293, 419, 420) or “substantial insights” (line 517), especially when, as you mentioned, your design is limited in certain points.
-Revise some of your conclusions. For instance, as far as I understand, the impact of ITV though plasticity is not corroborated with your study. Plasticity could have a genetic basis, being both plastic and genetic changes able to contribute to a phenotypic trend. However, inferences regarding phenotypic plasticity should be also supported by a specific set of evidentiary criteria (e.g. common garden experiment) and not only as an evidence from a null hypothesis, as seems you are doing here (see for instance Merila and Hendry 2014). Also, in line 528, I will limit it to abiotic and environmental changes instead of referring to “global changes”. So in these cases, please, tone down the message.
-I still consider that, using neutral markers the authors cannot infer a causal relationship with the phenotypic variation, so I recommend they should highlight this limitation too.
-You have mentioned in your responses and in line 222-224 that “We marked an area, ca. 6 m × 8 m, which corresponds to the smaller size of the exclosures” but it is not clear to me that you used this and not the 8 m x 12 m exclosures to analyse your data.
-I am still not convinced with the response around that “specific leaf area and leaf dry matter content was not measured in the standard way”. Please, explain the method used. In this regard, although understandable, I don´t feel the justification for the leaf dry matter is enough for a scientific work. Of course I understand that the field work frequently implies many limitations, but they should be considered to choose the design and measures to make. Considering this, I do not find a remarkable problem of using a representative subsample instead of all the samples, but it is when using unreported/untested method (which I assume if you mention “not measured in standard way”). Please, describe the method or limit this trait in your study. Also in this regard, I don´t feel “because they are easier to measure in the field” is an appropriate justification for a scientific work. The measure should be adequate to draw certain conclusions.
-You exposed in line 89-93 that large herbivores generally consume tall plants, and thus promote short plants because short plants are generally more nutritious, and it seems contradictory to me. Why large herbivores should select for taller plants rather than young and smaller plants if the larger ones are less palatable (e.g. more lignin content, lower energy assimilation, etc.)?
-In line 328, you say that “We checked residuals for each model, in all cases, models fitted the data reasonably well”. First, what “reasonable well” means? Do you refer that you visually checked that? Second, you should describe in detail the assumptions checked and the test used for that. For instance, did you used the Shapiro-Wilk normality test for checking the Normality of residuals or the Bartlett test for checking the homoscedasticity of variance? If your models did not comply these assumptions, you should apply adequate models (e.g. transformations, weighted least square regression) or use alternative ones.
-You frequently mention (e.g. 418-423, 442, …) that herbivores impact means, but you should also mention if responses/traits are negative or positively affected.
-In Line 417, I would change “can impact” by “could impact”
-In line 426-427, I don´t understand what “relative to ITV” means. Also, form the next lines, do you mean that the plant community structure may affect your results?

---

## Round 0.3 · accepted · Accept

Thank you very much for the thorough revision and the consideration of the comments of reviewer 3. I look forward to seeing your manuscript published.